# Synthesis and Biological Evaluation of Novel Ramalin Derivatives as Multi-Target Agents for Alzheimer’s Disease

**DOI:** 10.3390/molecules30092030

**Published:** 2025-05-02

**Authors:** Tai Kyoung Kim, Ju-Mi Hong, Yongeun Cho, Yeji Jeon, Heewon Cho, Jeongmi Lee, Jaewon Kim, Kyung Hee Kim, Il-Chan Kim, Se Jong Han, Hyuncheol Oh, Dong-Gyu Jo, Joung Han Yim

**Affiliations:** 1CRYOTECH Inc., 2F-211-3, 71 Mieumsandan 5-ro 41beon-gil, Gangseo-gu, Busan 46744, Republic of Korea; tkkim@cryotech.co.kr; 2Division of Polar Life Sciences, Korea Polar Research Institute, Incheon 21990, Republic of Korea; wnal5555@kopri.re.kr (J.-M.H.); ashcercle@kopri.re.kr (J.K.); kh313@kopri.re.kr (K.H.K.); ickim@kopri.re.kr (I.-C.K.); hansj@kopri.re.kr (S.J.H.); 3Department of Marine Sciences, Incheon National University, Incheon 22012, Republic of Korea; 4School of Pharmacy, Sungkyunkwan University, Suwon 16419, Republic of Korea; okcho9307@naver.com (Y.C.); jeonyeji183@gmail.com (Y.J.); hwcho1012@gmail.com (H.C.); jungmileedy@naver.com (J.L.); 5Department of Plant Biotechnology, Korea University, Seoul 02841, Republic of Korea; 6College of Pharmacy, Wonkwang University, Iksan 54538, Republic of Korea; hoh@wku.ac.kr; 7Samsung Advanced Institute for Health Science and Technology, Sungkyunkwan University, Seoul 06351, Republic of Korea; 8Biomedical Institute for Convergence, Sungkyunkwan University, Suwon 16419, Republic of Korea; 9Institute of Quantum Biophysics, Sungkyunkwan University, Suwon 16419, Republic of Korea

**Keywords:** Alzheimer’s disease, Ramalin derivatives, antioxidant activity, anti-inflammatory activity, BACE-1 inhibition, tau aggregation, multi-target therapy

## Abstract

Alzheimer’s disease (AD) is a complex neurodegenerative disorder characterized by cognitive decline, oxidative stress, neuroinflammation, amyloid-beta (Aβ) accumulation, and tau protein hyperphosphorylation. In this study, we synthesized novel Ramalin derivatives and evaluated their therapeutic potential against AD, focusing on antioxidant, anti-inflammatory, and neuroprotective activities. RA-2OMe, RA-4OMe, RA-2CF3, and RA-4OCF3 showed strong antioxidant effects, while RA-2OMe exhibited potent NO and NLRP3 inhibition (~20%). RA-NAP, RA-PYD, and RA-2Q showed moderate anti-inflammatory activity. BACE-1 inhibition was significant in RA-3CF3, RA-NAP, and RA-PYD, with IC_50_ values lower than that of positive control, indicating greater inhibitory potency. RA-NAP and RA-PYD effectively inhibited both Aβ and tau aggregation, highlighting their multi-target potential for AD therapy. These findings indicate that Ramalin derivatives exhibit potential for multi-target activity in AD treatment. However, further studies on their pharmacokinetics, in vivo efficacy, and long-term safety are required to confirm their therapeutic applicability.

## 1. Introduction

Alzheimer’s disease (AD) is the most prevalent neurodegenerative disorder worldwide, primarily affecting the elderly population and presenting a significant global health burden due to its increasing prevalence in aging societies [1,2]. AD is characterized by progressive memory decline, cognitive impairment, and neuronal degeneration, leading to severe functional deficits that ultimately compromise patients’ independence [3]. The pathophysiology of AD is highly complex and involves multiple interrelated pathological mechanisms, including amyloid-beta (Aβ) aggregation, tau pathology, oxidative stress, and neuroinflammation [4,5,6]. These interconnected pathological processes contribute to disease progression, making the development of effective treatment strategies particularly challenging [7,8].

Traditional single-target therapies, such as cholinesterase inhibitors and NMDA (*N*-methyl-*D*-aspartate) receptor antagonists, have been used to temporarily alleviate cognitive symptoms [9,10,11]; however, they do not halt or reverse the underlying neurodegenerative process [8]. BACE-1 (β-site amyloid precursor protein cleaving enzyme 1) inhibitors have been explored as potential AD therapeutics due to their ability to suppress Aβ production by preventing the cleavage of amyloid precursor protein (APP) into β-fragments, which are subsequently processed by γ-secretase to form Aβ [12]. Aβ activates microglia, triggering neuroinflammation, reducing synaptic function, and ultimately leading to neuronal death and cognitive decline [13]. Given these pathological implications, BACE-1 inhibition is considered a critical target in AD therapy, as it not only suppresses Aβ production but also reduces amyloid plaque formation and provides neuroprotective effects [14,15]. However, their clinical efficacy has been limited by toxicity concerns and incomplete disease modification [16,17]. The most commonly reported toxic effects of BACE-1 inhibitors include cognitive worsening, due to off-target inhibition affecting synaptic function, as well as hepatotoxicity and increased risk of infections, which have contributed to the failure of several candidates in clinical trials [17,18,19].

Additionally, accumulating evidence suggests that Aβ aggregation is closely linked to tau hyperphosphorylation, which further exacerbates neuronal damage by promoting the formation of neurofibrillary tangles [20]. Tau protein, which plays a crucial role in microtubule stabilization, becomes abnormally phosphorylated, losing its ability to bind microtubules properly. This leads to the formation of neurofibrillary tangles, which disrupt cytoskeletal integrity and impair neuronal function, ultimately contributing to neurodegeneration [21,22].

Given these challenges, recent advances in AD research emphasize the need for multi-target drug design strategies that simultaneously address Aβ accumulation, tau pathology, oxidative stress, and neuroinflammation [23,24,25,26,27]. To contribute to the development of **multi-target AD therapeutics**, this study focuses on designing and synthesizing novel **Ramalin derivatives** with enhanced pharmacological potential. Compared to combining multiple drugs, a single multi-target molecule offers advantages such as improved pharmacokinetic properties, reduced drug–drug interactions, and enhanced patient compliance, making it a more efficient therapeutic strategy for addressing the complex pathology of AD [28,29]. Inspired by recent advances in hybrid pharmacophore-based multi-target approaches, we hypothesize that rational structural modifications can enhance the neuroprotective properties of Ramalin derivatives [27,30,31,32].

Ramalin is a naturally occurring secondary metabolite isolated from the Antarctic lichen *Ramalina terebrata*, and it features a unique structure comprising a glutamic acid backbone and a hydrazinophenyl moiety [33]. This relatively simple molecular framework—consisting of a glutamic acid core, a hydrazine linker, and a phenyl ring—allows for diverse chemical modifications through the introduction of various functional groups. Such structural flexibility has been shown to significantly impact biological activities, including antioxidant, anti-inflammatory, BACE-1 inhibitory, and anti-tau aggregation effects. These findings suggest that even subtle changes to the functional groups can lead to substantial differences in biological responses, highlighting Ramalin as a chemically versatile and pharmacologically attractive scaffold for multi-target drug development. Previous studies have demonstrated its ability to scavenge free radicals, suppress microglial activation, and alleviate oxidative stress, thereby contributing to its neuroprotective properties [34]. Given these antioxidant and anti-inflammatory activities, Ramalin has emerged as a promising pharmacophore for the development of multi-target agents against neurodegenerative diseases such as Alzheimer’s disease, and serves as a valuable scaffold for structural optimization [35,36,37].

We hypothesize that **modifying the phenyl group of Ramalin or introducing functional groups such as methoxyl (-OCH_3_), trifluoromethyl (-CF_3_), and trifluoromethoxy (-OCF_3_) could enhance its therapeutic activity** (Figure 1). These substituents were selected due to their unique physicochemical properties: methoxyl groups, while nonpolar, share structural similarities with hydroxyl groups and may modulate hydrogen bonding interactions, whereas trifluoromethyl and trifluoromethoxy groups contribute hydrophobicity and strong electron-withdrawing effects, potentially enhancing biological activity. In our previous studies, we observed that biological activity varied significantly depending on the type and position of the substituent rather than simple differences in electronegativity [35,36,37]. Accordingly, in this study, we systematically evaluated how structural modifications—including various substituents and changes to aromatic (e.g., naphthalene, pyridine, quinoline) or non-aromatic alkyl groups—affect antioxidant, anti-inflammatory, BACE-1 inhibitory, and anti-tau aggregation activities (Figure 1).

## 2. Results

### 2.1. Synthesis of the Ramalin Derivatives

The Ramalin derivatives were synthesized using a previously established method [35,37]. The starting material, (S)-5-(benzyloxy)-4-(((benzyloxy)carbonyl)amino)-5-oxopentanoic acid (p-Glu), was dissolved in dichloromethane (DCM) and reacted with ethyl chloroformate (ECF) at 0 °C to form p-Glu-anhydride. To this activated intermediate, triethylamine (TEA) and hydrazine-substituted derivatives were added at the same temperature (0 °C), and the mixture was stirred at room temperature for 16 h to afford p-Glu-Hyd. The synthesized p-Glu-Hyd derivatives were purified by recrystallization using ethyl acetate (AcOEt) and hexane in a 1:3 ratio.

The benzyl (Bn) and benzyloxycarbonyl (Cbz) protecting groups were removed by catalytic hydrogenation using palladium on carbon (Pd/C, 10%) in methanol to obtain the final derivatives (Figure 1). However, for RA-NAP, which exhibits low solubility in methanol due to its high hydrophobicity, the deprotection reaction required precise adjustment of the methanol concentration. While standard deprotection reactions for benzyl and Cbz groups typically use 20–30 mM methanol at room temperature, the reaction for RA-NAP was conducted under highly diluted conditions (approximately 2 mM) for an extended reaction time (e.g., 24 h) to ensure successful deprotection.

### 2.2. Antioxidant Effect of the Ramalin Derivatives

The antioxidant activities of the Ramalin derivatives were measured using the 2,2-diphenyl-1-picrylhydrazyl (DPPH) radical scavenging assay, with absorbance recorded at 540 nm. The IC_50_ values of the derivatives were determined in methanol, and the results are as follows: RA-2OMe (2.81 ± 0.03 µM), RA-4OMe (5.49 ± 0.63 µM), RA-2CF3 (3.0 ± 0.03 µM), RA-3CF3 (31.68 ± 14.61 µM), RA-4OCF3 (3.77 ± 0.09 µM), RA-NAP (1.77 ± 0.12 µM), RA-PYD (7.4 ± 3.54 µM), RA-2Q (9.7 ± 0.32 µM), RA-DMe (598.35 ± 143.78 µM), RA-IPr (2580.01 ± 359.3 µM), and RA-Morp (4800.01 ± 777.82 µM) (Table 1).

Derivatives such as RA-2OMe, RA-2CF3, RA-4OCF3, and RA-NAP exhibited DPPH IC_50_ values comparable to that of Ramalin (IC_50_ = 1.25 ± 0.01 µM), suggesting similar antioxidant effects. On the other hand, RA-3CF3 showed significantly weaker antioxidant activity compared to its 2- and 4-substituted counterparts, despite having the same trifluoromethyl (-CF_3_) functional group. This suggests that the trifluoromethyl group at the 3-position may have a structural or electronic effect that reduces antioxidant efficiency, highlighting the importance of substitution position in modulating radical scavenging activity.

Furthermore, RA-DMe, RA-IPr, and RA-Morp displayed significantly reduced antioxidant effects, with IC_50_ values increasing to the millimolar range, suggesting that steric hindrance or loss of electronic conjugation may negatively impact activity. In contrast, derivatives lacking a phenyl ring, such as RA-NAP, RA-PYD, and RA-2Q, retained moderate antioxidant activity, likely due to their ability to participate in electronic resonance through alternative conjugation pathways. This finding supports the hypothesis that the electronic interaction between the hydrazine (-NH-NH_2_) moiety and the aromatic or conjugated system plays a crucial role in determining antioxidant activity.

### 2.3. Blood–Brain Barrier Permeability Evaluation

Table 1 presents the evaluation of the blood–brain barrier (BBB) permeability and antioxidant effects of Ramalin and its derivatives. The molecular properties were determined using the SwissADME platform (www.swissadme.ch) [16], including their total polar surface areas (TPSAs), isocratic log partition coefficients (iLogPs), X-ray log partition coefficients (XLogPs), and molecular weights (MWs). The TPSA values of these derivatives ranged from 95.66 to 117.34 Å^2^, with Ramalin having a TPSA of 128 Å^2^. Higher TPSA values typically indicate reduced BBB permeability. The iLogP and XLogP values reflect the lipophilicity of the compounds. Most of the derivatives show values that suggest low lipophilicity, which further limits their ability to cross the BBB. Based on the results obtained using the SwissADME platform, all the derivatives were marked as “No” for BBB permeability. The MW of the compounds ranged from 189.21 to 321.26 g/mol, with that of Ramalin being 253.26 g/mol. While a low molecular weight is typically favorable for BBB penetration, high TPSA and unfavorable lipophilicity indicate that these derivatives are unlikely to efficiently penetrate the BBB [38]. This result aligns with the observation that nearly 98% of small neurotherapeutic molecules are incapable of crossing the BBB [39].

Strategies such as prodrug design and the use of nanoparticles or liposomes may be useful in improving the BBB permeability [38,40]. Other studies have reported the use of these approaches for enhancing the BBB permeability of compounds with similar challenges [41,42]. It is worth noting that, while these in silico predictions provide valuable insights, further experimental validation using methods such as the PAMPA-BBB assay is necessary to confirm the actual BBB permeability of these compounds [43].

### 2.4. Anti-Inflammatory Effect of Ramalin Derivatives

#### 2.4.1. Effect of Ramalin Derivatives on Cell Viability

The toxicities of the synthesized Ramalin derivatives against the RAW 264.7 cell line were assessed using the MTT assay [44]. RAW 264.7 cells, a murine macrophage cell line, were chosen for cytotoxicity evaluation due to their well-characterized immune response and widespread use in assessing the biocompatibility of small molecules, particularly those with potential anti-inflammatory properties. These cells provide a reliable in vitro model for screening cytotoxic effects before proceeding to further biological assays [45]. RAW 264.7 cells were seeded at a density of 2 × 10^5^ cells per well in a 96-well plate and incubated in DMEM supplemented with 10% fetal bovine serum (FBS) and 1% penicillin/streptomycin under 5% CO_2_ at 37 °C. The cells were treated with the derivatives at concentrations of 0, 5, 10, 20, and 40 μM and incubated for 24 h. MTT solution (0.5 mg/mL) was added to each well, and after a 4 h incubation, formazan crystals were dissolved in DMSO. Absorbance was measured at 570 nm using a microplate reader. The MTT assay results showed that Ramalin exhibited a concentration-dependent reduction in cell viability, with viability decreasing to approximately 60~70% at higher concentrations (20~40 μM). This suggests that Ramalin itself exhibited moderate cytotoxicity at higher doses, unlike some of its derivatives that maintained cell viability above 100%. These findings highlight that structural modifications to Ramalin may contribute to reducing cytotoxicity while maintaining biological activity. The results demonstrated that all Ramalin derivatives did not exhibit significant cytotoxic effects up to 40 μM, as indicated by cell viability rates that remained near or above 100% across all tested concentrations (Figure 2). Specifically, RA-2OMe, RA-4OMe, RA-2CF3, RA-3CF3, and RA-4OCF3 exhibited cell viability above 100%, suggesting no apparent cytotoxicity. RA-PYD, RA-2Q, RA-IPr, and RA-Morp also showed high viability, with no significant reduction in cell survival at tested concentrations.

However, at higher concentrations (≥40 μM), RA-NAP and RA-DMe displayed slightly reduced cell viability, indicating a potential concentration-dependent cytotoxic effect for these compounds. Nonetheless, even at 40 μM, the viability remained above 80%, suggesting relatively low toxicity. Based on these findings, a concentration threshold of 40 μM was established as the upper limit for safe usage in subsequent experiments. All subsequent biological assays were conducted at concentrations of 40 μM or lower to ensure non-cytotoxic conditions.

#### 2.4.2. Effects of Ramalin Derivatives on Nitric Oxide Production

To evaluate the anti-inflammatory activity of Ramalin derivatives, we measured nitric oxide (NO) production in LPS-stimulated RAW 264.7 cells. The cells were pre-treated with various concentrations (5, 10, 20, and 40 µM) of Ramalin derivatives for 1 h, followed by stimulation with lipopolysaccharide (LPS, 1 µg/mL) for 24 h. NO levels in the culture supernatants were quantified using the Griess reagent assay, and absorbance was measured at 540 nm.

The results showed that NO production was significantly increased upon LPS stimulation compared to the untreated control group (CON), confirming the inflammatory response (Figure 3). Ramalin treatment also resulted in a concentration-dependent reduction of NO levels, with significant inhibition observed at 20 and 40 µM. At 40 µM, Ramalin reduced NO production by approximately 50% relative to the LPS-treated group, indicating a strong anti-inflammatory effect. This finding indicates that the parent compound itself exhibits notable NO inhibition, which may serve as a basis for the observed effects in its derivatives.

However, treatment with derivatives structurally similar to Ramalin, including RA-2OMe, RA-4OMe, RA-3CF3, and RA-4OCF3, resulted in a concentration-dependent reduction in NO levels. Among these, RA-4OMe and RA-2OMe exhibited the most significant NO inhibition, reducing NO production by approximately 25~35% at 40 µM. In contrast, RA-2CF3, RA-NAP, and RA-DMe showed a moderate NO-reducing effect, but their inhibition was not clearly concentration-dependent. Other derivatives, such as RA-PYD, RA-2Q, RA-IPr, and RA-Morp, did not significantly suppress NO production across the tested concentrations. These findings suggest that structural similarities with Ramalin, particularly the presence of specific functional groups, may enhance NO-inhibitory activity.

#### 2.4.3. Effect of Ramalin Derivatives on NLRP3 Level

In LPS-stimulated RAW 264.7 cells, the expression level of NLRP3 was significantly high, indicating the activation of the inflammasome pathway. For evaluating the NLRP3 inhibitory activity, the highest concentration of 40 µM was selected, as this concentration had previously been confirmed to maintain cell viability (Section 2.4.1).

Ramalin itself exhibited minimal NLRP3 inhibition, with an inhibition rate of approximately 5%, suggesting a weak effect on inflammasome suppression. This suggests that while Ramalin possesses antioxidant properties, its direct impact on NLRP3 activation is limited. These findings highlight that structural modifications in Ramalin derivatives may enhance their ability to modulate neuroinflammation by improving NLRP3 inhibition.

As shown in Figure 4, several Ramalin derivatives inhibited NLRP3 activation. Specifically, derivatives such as RA-2OMe, RA-4OMe, RA-2CF3, RA-4OCF3, RA-PYD, RA-2Q, and RA-Morp showed notable inhibitory activities, with inhibition rates of approximately 20% or higher. Among these, RA-2OMe demonstrated the most potent inhibitory effect, resulting in the greatest reduction in NLRP3 levels.

In contrast, RA-3CF3 and RA-NAP exhibited relatively low NLRP3-inhibitory activities, with inhibition levels of 10–15%, whereas RA-DMe and RA-IPr showed no significant NLRP3-inhibitory effects. These results suggested that NLRP3 inhibition may be partially associated with the antioxidant activity. However, an interesting observation was noted with RA-Morp, which demonstrated NLRP3-inhibitory activity despite its lack of antioxidant effects. This unique finding indicates that NLRP3 inhibition may involve additional pathways beyond simple antioxidant mechanisms.

Overall, these results highlight the potential of certain Ramalin derivatives in modulating neuroinflammatory responses through NLRP3 inhibition, which could be advantageous for developing therapeutics targeting neurodegenerative diseases like AD.

### 2.5. BACE-1 Inhibitory Activities of Ramalin Derivatives

The BACE-1 inhibitory activities of the synthesized Ramalin and its derivatives were evaluated using a fluorescence resonance energy transfer (FRET)-based assay, and their IC_50_ values are summarized in Table 2. The results represent the mean ± standard deviation of at least three independent experiments, with LY2811376 used as a standard positive control agent. Ramalin itself exhibited a moderate BACE-1 inhibitory effect, with an IC_50_ value of 2.71 ± 0.49 μM. Although its activity was weaker than the most potent derivatives, such as RA-3CF3 and RA-PYD, it demonstrated greater inhibitory potential than the positive control LY28113761. This suggests that Ramalin serves as a valuable scaffold for further structural modifications to enhance BACE-1 inhibition.

Among the tested compounds, RA-3CF3 (IC_50_ = 0.33 ± 0.79 μM), RA-PYD (IC_50_ = 0.32 ± 0.08 μM), and RA-NAP (IC_50_ = 0.66 ± 1.05 μM) exhibited the most potent inhibitory activities, showing IC_50_ values approximately 10 times lower than that of Ramalin. These results indicate significantly improved inhibitory potential compared to Ramalin itself.

Moderate BACE-1 inhibitory effects were observed for RA-2OMe (IC_50_ = 7.97 ± 2.46 μM), RA-2CF3 (IC_50_ = 11.56 ± 14 μM), RA-4OCF3 (IC_50_ = 9.89 ± 13.59 μM), and RA-4OMe (IC_50_ = 24.99 ± 5.72 μM), while RA-Morp exhibited weaker activity with an IC_50_ of 16.06 ± 11.89 μM. Notably, RA-2Q, RA-DMe, and RA-IPr showed no detectable BACE-1 inhibitory activity (ND).

Compared to previous studies, where the best-performing Ramalin derivatives exhibited IC_50_ values approximately two times lower than Ramalin [35], the current study highlights the remarkable improvement of certain newly synthesized derivatives, particularly RA-3CF3 and RA-PYD, which exhibit IC_50_ values nearly ten times lower than Ramalin. These results suggest that the strong BACE-1 inhibitory activity observed in certain structures is influenced not only by the functional groups but also by the structural characteristics that affect the binding affinity with BACE-1. Notably, RA-3CF3 and RA-PYD exhibited potent inhibitory activities, which may be attributed to their specific spatial arrangements and electronic properties that enhance their interactions with the BACE-1 active site. In contrast, derivatives such as RA-DMe and RA-IPr, which contain bulky alkyl groups, showed no detectable inhibitory activity, indicating that steric hindrance from these groups may interfere with the effective binding to BACE-1. Overall, these findings highlight the critical role of structural features in BACE-1 inhibition and support the potential of certain derivatives as promising BACE-1-targeting candidates in the therapy for AD.

### 2.6. Anti-Tau Aggregation Effects of Ramalin Derivatives

Neurofibrillary tangles, composed of hyperphosphorylated tau protein, represent a key hallmark of AD pathology [46]. To investigate the anti-tau aggregation effects of Ramalin derivatives, we employed a bimolecular fluorescence complementation (BiFC) system using tau protein (Figure 5A) [36]. A tau-BiFC SH-SY5Y cell line was established to assess the inhibitory effects of Ramalin derivatives on tau aggregation. This system enables the detection of tau–tau interactions through enhanced green fluorescent protein (EGFP) fluorescence when tau aggregation occurs [47].

As shown in Figure 5B, treatment with Ramalin significantly reduced tau aggregation compared to the control group (**** *p* < 0.0001), confirming its strong anti-tau aggregation activity. Among the Ramalin derivatives, RA-NAP, RA-PYD, and RA-2Q exhibited significant reductions in tau aggregation (*** *p* < 0.001, * *p* < 0.05), suggesting their potential neuroprotective effects. While RA-2OMe and RA-4OMe showed moderate inhibitory effects, other derivatives such as RA-DMe, RA-IPr, and RA-Morp did not exhibit significant anti-tau aggregation activity.

These findings indicate that specific structural modifications, particularly those involving naphthalene (NAP), pyridine (PYD), and quinoline (2Q) groups, may enhance the ability of Ramalin derivatives to inhibit tau aggregation. These findings highlight the potential of these derivatives as promising candidates for targeting tau pathology in AD therapy.

## 3. Discussion

AD is a multifactorial neurodegenerative disorder characterized by cognitive decline, neuroinflammation, oxidative stress, Aβ accumulation, and tau protein hyperphosphorylation [48,49]. In this study, we comprehensively evaluated the antioxidant, anti-inflammatory, and neuroprotective activities of novel Ramalin derivatives to explore their therapeutic potential.

The antioxidant activities of Ramalin derivatives, assessed using the DPPH radical scavenging assay, demonstrated that the IC_50_ values of RA-2OMe, RA-4OMe, RA-2CF3, and RA-4OCF3 were comparable to those of Ramalin, indicating their strong radical scavenging capabilities. In contrast, derivatives such as RA-DMe, RA-IPr, and RA-Morp showed significantly reduced antioxidant effects, suggesting that their structural modifications affect the electronic resonance and that hydrazine–phenyl interactions play a critical role in maintaining antioxidant activity. These findings are consistent with those of previous studies emphasizing the importance of electron-donating groups in enhancing antioxidant potential [35].

The predicted BBB permeability, analyzed via SwissADME, revealed unfavorable characteristics due to high TPSA and low lipophilicity. Despite their low molecular weights, all derivatives were predicted to have poor BBB permeability, aligning with the challenge that most small neurotherapeutic molecules face in crossing the BBB [38,42]. Strategies such as prodrug design, nanoparticle conjugation, or liposomal delivery may improve BBB permeability [42,50].

In the NO inhibition assay using LPS-stimulated RAW 264.7 cells, derivatives structurally similar to Ramalin RA-2OMe, RA-4OMe, RA-2CF3, RA-3CF3, and RA-4OCF3 showed concentration-dependent NO reduction. This suggests that methoxy and trifluoromethyl substitutions may enhance anti-inflammatory potential by modulating oxidative stress pathways. Moreover, the structural similarity of these derivatives, where a hydrazinophenyl group is linked to glutamic acid, supports the hypothesis that this pharmacophore is crucial for Ramalin’s anti-AD activity.

Interestingly, RA-NAP and RA-2Q also exhibited moderate NO-inhibitory effects, likely due to their electronic properties facilitating resonance with hydrazine, as seen with naphthalene and quinoline moieties. The NLRP3 inflammasome inhibition assay showed similar trends to NO inhibition, with RA-2OMe, RA-4OMe, RA-2CF3, RA-3CF3, and RA-4OCF3 displaying inhibitory activity, and RA-2OMe demonstrating the highest activity (~20% inhibition). Additionally, RA-NAP, RA-PYD, and RA-2Q exhibited comparable inhibitory effects, while RA-Morp was the only alkyl-substituted derivative to show NLRP3 inhibitory activity. Although not all derivatives exhibited strong anti-inflammatory effects, those containing a hydrazinophenyl moiety—such as RA-2OMe, RA-4OMe, RA-2CF3, RA-3CF3, and RA-4OCF3—consistently demonstrated moderate to strong activity in both NO and NLRP3 inhibition assays, likely due to their structural similarity to the parent compound Ramalin [35,36].

The BACE-1 inhibitory assay revealed that RA-3CF3, RA-NAP, and RA-PYD exhibited strong inhibition, with IC_50_ values significantly lower than the positive control LY28113761. Notably, compared to previous studies, where the best-performing Ramalin derivatives exhibited only moderate improvements, the newly synthesized derivatives demonstrated up to tenfold greater inhibitory potency than Ramalin itself. This suggests that structural factors, rather than the electronic nature of substituents, play a crucial role in enhancing BACE-1 binding affinity. However, no clear structural correlations were identified, highlighting the need for further molecular docking studies to elucidate the precise binding mechanisms.

Furthermore, **anti-tau aggregation assays** demonstrated that **Ramalin, RA-2OMe, RA-NAP, RA-PYD, and RA-2Q effectively inhibited tau fibril formation**. Interestingly, **despite strong BACE-1 inhibitory activity, RA-3CF3 did not show significant anti-tau effects**, suggesting that **BACE-1 inhibition alone may not directly correlate with tau aggregation suppression** [19,51]. Notably, **RA-PYD and RA-NAP exhibited comprehensive biological activities, including antioxidant, anti-inflammatory, BACE-1 inhibitory, and anti-tau aggregation effects**, highlighting their potential as **multi-target agents for AD therapy**.

Collectively, these findings suggest that Ramalin derivatives exhibit multifunctional properties relevant to AD therapy, modulating oxidative stress, neuroinflammation, tau aggregation, and BACE-1 activity. The observed differences in experimental concentrations across assays reflect the distinct biological mechanisms being evaluated. Specifically, NLRP3 inhibition required higher concentrations (40 µM) due to the robustness of inflammasome activation, while tau aggregation assays were conducted at 10 µM to minimize potential non-specific effects. In contrast, BACE-1 inhibition was observed in the low micromolar range, aligning with enzyme kinetics. These findings highlight the multifunctional nature of these compounds while emphasizing the need for further dose-dependent in vivo validation to optimize their therapeutic potential. However, their therapeutic relevance requires further in vivo validation and structural optimization to enhance efficacy and bioavailability. Structural modifications preserved or even enhanced biological activities in some cases, suggesting that strategic modifications could optimize their therapeutic efficacy. Among the tested derivatives, RA-PYD and RA-NAP exhibited the most promising multi-target activities, demonstrating potent antioxidant, anti-inflammatory, BACE-1 inhibitory, and anti-tau aggregation effects. Building on these results, future studies will focus on synthesizing novel derivatives featuring structurally similar functional groups to RA-PYD and RA-NAP, aiming to identify derivatives with enhanced bioactivity. Additionally, given the current limitations in BBB permeability, we plan to design structures optimized for improved BBB penetration while maintaining or enhancing biological activity.

Future research should also focus on in vivo validation, as well as computational modeling approaches to explore molecular interactions, which will provide valuable insights into structure–activity relationships (SAR) and guide the rational development of more potent and BBB-permeable Ramalin-based AD therapeutics. These efforts will contribute to overcoming current limitations and advancing the development of effective multi-target drugs for AD therapy.

Detailed experimental data and characterization of the synthesized derivatives are provided in the Appendix A.

## 4. Materials and Methods

### 4.1. General Experimental Information

All solvents and reagents were purchased from commercial sources, including Merck (Darmstadt, Germany) and TCI (Tokyo, Japan), and were used without further purification. All glassware was thoroughly dried in a 60 °C drying oven or flamed and immediately cooled under a dry argon stream before use. Filters were acquired from GE Healthcare (GF/F, 0.7 µm, Whatman, UK). All reactions were conducted under an inert argon atmosphere. Solvents and liquid reagents were loaded into syringes prior to use. Organic extracts were dried over Na_2_SO_4_ and concentrated under reduced pressure using a rotary evaporator (Eyela, Tokyo, Japan). Any residual solvents were eliminated under high vacuum (Vacuubrand RZ 2.5, Wertheim, Germany, 1 × 10⁻^2^ mbar). Purification was carried out using a Yamazen Smart Flash (EPCLC AI-580S) medium-pressure liquid chromatography (MPLC) system. Accurate mass spectra were recorded using an AB Sciex Triple TOF 4600 (Framingham, MA, USA) in direct injection mode. Nuclear magnetic resonance (NMR) spectra were obtained on a Jeol JNM ECP-400 spectrometer (Jeol Ltd., Tokyo, Japan) with either a D_2_O-acetone-d_6_ mixture (6:1 *v*/*v*) containing 0.01 mg/mL sodium trimethylsilylpropanesulfonate (DSS) or dimethyl sulfoxide (DMSO)-d_6_ as solvents. Chemical shifts were referenced using the internal standard or residual solvent signals [D_2_O (with DSS)-acetone-d_6_: δH 0.00/δC 29.8, DMSO-d_6_: δH 2.50/δC 39.5]. Peak splitting patterns were denoted as follows: m (multiplet), s (singlet), d (doublet), t (triplet), q (quartet), sep (septet), dd (doublet of doublets), td (triplet of doublets), dt (doublet of triplets), ddd (doublet of doublets of doublets), and br d (broad doublet). Absorbance measurements were performed using a microplate reader (Thermo Scientific Inc., San Diego, CA, USA) and a multi-mode plate reader (Multiskan™ GO, Thermo Scientific, Waltham, MA, USA).

### 4.2. Synthesis and Characterization

#### 4.2.1. General Method for the Synthesis of p-Glu-Hyd Analogues

A 250 mL round flask equipped with a magnetic stir bar was charged with 1-Benzyl-N-Cbz-L-glutamic acid (p-Glu, 3.0 g, 8.08 mmol) dissolved in 50 mL of DCM. The reaction mixture was cooled to 0 °C, and TEA (1.2 eq, 9.70 mmol, 1352 µL) was added dropwise. After stirring for 10 min, ECF (1.2 eq, 9.70 mmol, 942 µL) was slowly introduced over the course of 1 h. The reaction was maintained at 0 °C with continuous stirring for an additional 4 h. Separately, hydrazine hydrochloride (1.2 eq, 9.70 mmol) was dissolved in TEA (1.2 eq, 9.70 mmol, 1352 µL) in a 100 mL pear-shaped flask. This solution was gradually added to the main reaction flask over 1 h while maintaining the temperature at 0 °C. Upon completion of the addition, the reaction mixture was allowed to reach room temperature and stirred for 16 h.

The reaction mixture was then sequentially washed with distilled water, 1 N HCl, 0.5 N NaHCO_3_, and finally, distilled water to ensure phase separation. The organic layer was collected, dried over Na_2_SO_4_, and concentrated under reduced pressure using a rotary evaporator. The crude product was purified via recrystallization from an ethyl acetate (AcOEt) and n-hexane (1:5) mixture.

#### 4.2.2. General Method for the Synthesis of RA Analogues

A 500 mL round-bottom flask containing a magnetic stir bar was charged with the appropriate p-Glu-Hyd analogue and 10 weight % palladium on carbon in 200 mL of MeOH. The reaction mixture was stirred under a hydrogen atmosphere (1 atm, hydrogen) balloon) for 16 h. After completion, the reaction mixture was filtered using glass microfiber filter paper (0.4 µm), and the filtrate was concentrated using a rotary evaporator. The crude product was purified by recrystallization from a MeOH/AcOEt (1:8) mixture.

*N^5^*-((2-methoxyphenyl)amino)glutamine (RA-2OMe). From (*S*)-5-(benzyloxy)-4-(((benzyloxy)carbonyl)amino)-5-oxopentanoic acid; 1.34 g, 62%, white solid; ^1^H NMR (400 MHz, DMSO-*d*_6_): *δ* 7.01 (m, 1H, PhH), 6.94 (m, 2H, PhH), 6.85 (m, 1H, PhH), 3.88 (s, 3H, OCH_3_) 3.80 (t, *J* = 6.4 Hz, 1H, H-2), 2.52 (dt, *J* = 7.4, 6.4 Hz, 2H, H-3), 2.19 (m, 2H, H-4); ^13^C NMR (100 MHz, DMSO-*d*_6_): *δ* 174.7, 173.9, 147.5, 136.5, 121.7, 121.6, 113.0, 111.8, 56.2, 54.5, 29.8, 26.3; HRESIMS *m*/*z* 268.1291 [M + H]+ (calcd for C_12_H_18_N_3_O_3_, 268.1297).

*N^5^*-((4-methoxyphenyl)amino)glutamine (RA-4OMe). From (*S*)-5-(benzyloxy)-4-(((benzyloxy)carbonyl)amino)-5-oxopentanoic acid; 1.32 g, 61%, white solid; ^1^H NMR (400 MHz, DMSO-*d*_6_): *δ* 6.94 (m, 2H, PhH), 6.88 (d, *J* = 9.3 Hz, 2H, PhH), 3.81 (t, *J* = 6.0 Hz, 1H, H-2), 3.78 (s, 3H, OCH_3_), 2.53 (td, *J* = 7.5, 3.9 Hz, 2H, H-3), 2.20 (m, 2H, H-4); ^13^C NMR (100 MHz, DMSO-*d*_6_): *δ* 174.7, 173.8, 154.0, 141.6, 115.5, 115.3, 56.0, 54.4, 32.4, 29.0; HRESIMS *m*/*z* 268.1296 [M + H]+ (calcd for C_12_H_18_N_3_O_3_, 268.1297)

*N^5^*-((2-(triflouromethyl)phenyl)amino)glutamine (RA-2CF3). From (*S*)-5-(benzyloxy)-4-(((benzyloxy)carbonyl)amino)-5-oxopentanoic acid; 1.48 g, 60%, white solid; ^1^H NMR (400 MHz, DMSO-*d*_6_): *δ* 7.60 (td, *J* = 6.6, 4.5 Hz, 1H, PhH), 7.51 (dd, *J* = 6.7, 1.7 Hz, 1H, PhH), 7.04 (d, *J* = 4.6 Hz, 1H, PhH), 7.02 (d, *J* = 4.6 Hz, 1H, PhH), 3.81 (t, *J* = 6.2 Hz, 1H, H-2), 2.58 (m, 2H, H-4), 2.20 (m, 2H, H-3); ^13^C NMR (100 MHz, DMSO-*d*_6_): *δ* 174.9, 173.9, 145.0, 134.0, 126.9 (q, *J_CF_* = 5.4 Hz, C-3′), 124.7 (q, *J_CF_* = 270.7 Hz, CF_3_), 120.4, 114.0 (q, *J_CF_* = 30.8 Hz, C-2′), 113.6, 54.4, 29.7, 26.3; HRESIMS *m*/*z* 306.1071 [M + H]+ (calcd for C_12_H_15_F_3_N_3_O_3_, 306.1065)

*N^5^*-((3-(triflouromethyl)phenyl)amino)glutamine (RA-3CF3). From (*S*)-5-(benzyloxy)-4-(((benzyloxy)carbonyl)amino)-5-oxopentanoic acid; 1.21 g, 49%, white solid; ^1^H NMR (400 MHz, DMSO-*d*_6_): *δ* 7.34 (t, *J* = 7.6 Hz, 1H, PhH), 6.99 (d, *J* = 7.6 Hz, 1H, PhH), 6.97 (d, *J* = 8.8 Hz, 1H, PhH), 6.95 (s, 1H, PhH), 3.24 (t, *J* = 6.4 Hz, 1H, H-2), 2.38 (m, 2H, H-4), 1.90 (m, 2H, H-3); ^13^C NMR (100 MHz, DMSO-*d*_6_): *δ* 172.4, 170.3, 150.7, 130.4, 130.2 (q, *J_CF_* = 30.6 Hz, C-3′), 125.0 (q, *J_CF_* = 270.7 Hz, CF_3_), 116.1, 114.9 (q, *J_CF_* = 3.1 Hz, C-4′), 108.4 (q, *J_CF_* = 3.8 Hz, C-2′), 54.1, 30.4, 27.6; HRESIMS *m*/*z* 306.1072 [M + H]+ (calcd for C_12_H_15_F_3_N_3_O_3_, 306.1065)

*N^5^*-((4-(triflouromethoxy)phenyl)amino)glutamine (RA-4OCF3). From (*S*)-5-(benzyloxy)-4-(((benzyloxy)carbonyl)amino)-5-oxopentanoic acid; 1.58 g, 61%, white solid; ^1^H NMR (400 MHz, DMSO-*d*_6_): *δ* 7.23 (br d, *J* = 8.8 Hz, 2H, PhH), 6.91 (d, *J* = 8.8 Hz, 2H, PhH), 3.81 (t, *J* = 6.2 Hz, 1H, H-2), 2.58 (m, 2H, H-4), 2.20 (m, 2H, H-3); ^13^C NMR (100 MHz, DMSO-*d*_6_): *δ* 175.4, 174.3, 146.7, 142.4, 122.4, 118.8 (q, *J_CF_* = 253.6 Hz, OCF_3_), 114.0, 54.0, 29.5, 26.2; HRESIMS *m*/*z* 322.1018 [M + H]+ (calcd for C_12_H_15_F_3_N_3_O_3_, 322.1015)

*N^5^*-(naphthalen-2-ylamino)glutamine (RA-NAP). From (*S*)-5-(benzyloxy)-4-(((benzyloxy)carbonyl)amino)-5-oxopentanoic acid; 0.81 g, 35%, white solid; ^1^H NMR (400 MHz, DMSO-*d*_6_): *δ* 8.18 (m, 1H, NAP-H), 7.80 (d, *J* = 9.2 Hz, 1H, NAP-H), 7.43 (m, 2H, NAP-H), 7.25 (m, 2H, NAP-H), 6.79 (d, *J* = 6.4 Hz, 1H, NAP-H), 3.83 (m, 1H, H-2), 2.04 (m, 2H, H-4), 1.95 (m, 2H, H-3); ^13^C NMR (100 MHz, DMSO-*d*_6_): *δ* 176.9, 171.8, 144.1, 133.8, 127.9, 126.3, 125.7, 124.4, 122.1, 121.7, 118.2, 104.9, 53.7, 29.8, 27.2; HRESIMS *m*/*z* 288.1338 [M + H]+ (calcd for C_15_H_18_N_3_O_3_, 288.1348)

*N^5^*-(pyridin-2-ylamino)glutamine (RA-PYD). From (*S*)-5-(benzyloxy)-4-(((benzyloxy)carbonyl)amino)-5-oxopentanoic acid; 1.44 g, 75%, white solid; ^1^H NMR (400 MHz, DMSO-*d*_6_): *δ* 8.11 (ddd, *J* = 8.9, 7.2, 1.6 Hz, 1H, Pyridin-H-3′), 8.00 (ddd, *J* = 6.4, 1.6, 0.8 Hz, 1H, Pyridin-H-5′), 7.23 (dt, *J* = 9.0, 1.0 Hz, 1H, Pyridin-H-4′), 7.16 (ddd, *J* = 7.3, 6.3, 1.1 Hz, 1H, Pyridin-H-6′), 3.82 (t, *J* = 6.0 Hz, 1H, H-2), 2.66 (m, 2H, H-4), 2.21 (m, 2H, H-3); ^13^C NMR (100 MHz, DMSO-*d*_6_): *δ* 175.3, 173.9, 153.7, 145.5, 136.3, 116.0, 111.9, 54.1, 29.8, 26.3; HRESIMS *m*/*z* 239.1179 [M + H]+ (calcd for C_10_H_15_N_4_O_3_, 239.1144)

*N^5^*-(quinolin-2-ylamino)glutamine (RA-2Q). From (*S*)-5-(benzyloxy)-4-(((benzyloxy)carbonyl)amino)-5-oxopentanoic acid; 1.65 g, 71%, yellow solid; ^1^H NMR (400 MHz, DMSO-*d*_6_): *δ* 8.13 (d, *J* = 9.2 Hz, 1H, Quinolin-H-7′), 7.77 (d, *J* = 8.4 Hz, 1H, Quinolin-H-3′), 7.70 (d, *J* = 8.4 Hz, 1H, Quinolin-H-6′), 7.60 (td, *J* = 7.6, 8.0 Hz, 1H, Quinolin-H-4′), 7.33 (td, *J* = 7.2, 7.6 Hz, 1H, Quinolin-H-5′), 7.02 (d, *J* = 9.2 Hz, 1H, Quinolin-H-8′), 3.75 (t, *J* = 6.4 Hz, 1H, H-2), 2.59 (m, 2H, H-4), 2.19 (m, 2H, H-3); ^13^C NMR (100 MHz, DMSO-*d*_6_): *δ* 174.5, 173.5, 156.3, 141.6, 141.6, 131.9, 128.7, 125.2, 123.9, 121.6, 110.9, 54.4, 30.2, 26.1; HRESIMS *m*/*z* 289.1307 [M + H]+ (calcd for C_14_H_17_N_4_O_3_, 289.1301)

*N^5^*-(dimethylamino)glutamine (RA-DMe). From (*S*)-5-(benzyloxy)-4-(((benzyloxy)carbonyl)amino)-5-oxopentanoic acid; 1.16 g, 76%, yellow solid; ^1^H NMR (400 MHz, DMSO-*d*_6_): *δ* 3.73 (m, 1H, H-2), 2.68 (s, 2H, H-1′, H-2′) 2.32 (m, 2H, H-4), 2.07 (m, 2H, H-3); ^13^C NMR (100 MHz, DMSO-*d*_6_): *δ* 173.1, 172.9, 53.9, 46.5, 30.0, 25.7; HRESIMS *m*/*z* 190.1175 [M + H]+ (calcd for C_7_H_16_N_3_O_3_, 190.1191)

*N^5^*-(Isopropylamino)glutamine (RA-IPr). From (*S*)-5-(benzyloxy)-4-(((benzyloxy)carbonyl)amino)-5-oxopentanoic acid; 1.18 g, 72%, yellow solid; ^1^H NMR (400 MHz, DMSO-*d*_6_): *δ* 3.68 (t, *J* = 6.0, 1H, H-2), 2.97 (sep, 1H, Ipr-H-1′) 2.37 (m, 2H, H-4), 2.11 (m, 2H, H-3), 0.95 (d, *J* = 6.4 Hz, 3H, Ipr-H-2′, H-3′); ^13^C NMR (100 MHz, DMSO-*d*_6_): *δ* 173.1, 172.9, 53.9, 46.5, 30.0, 25.7; HRESIMS *m*/*z* 204.1330 [M + H]+ (calcd for C_8_H_18_N_3_O_3_, 204.1348)

*N^5^*-(Isopropylamino)glutamine (RA-Morp). From (*S*)-5-(benzyloxy)-4-(((benzyloxy)carbonyl)amino)-5-oxopentanoic acid; 1.47 g, 79%, yellow solid; ^1^H NMR (400 MHz, DMSO-*d*_6_): *δ* 3.68 (t, *J* = 6.0, 1H, H-2), 2.97 (sep, 1H, Ipr-H-1′) 2.37 (m, 2H, H-4), 2.11 (m, 2H, H-3), 0.95 (d, *J* = 6.4 Hz, 3H, Ipr-H-2′, H-3′); ^13^C NMR (100 MHz, DMSO-*d*_6_): *δ* 173.1, 172.9, 53.9, 46.5, 30.0, 25.7; HRESIMS *m*/*z* 232.1279 [M + H]+ (calcd for C_9_H_18_N_3_O_4_, 232.1297)

### 4.3. Antioxidant Activity Assay

The antioxidant properties of Ramalin and its chloride derivatives were assessed through their radical scavenging abilities using the DPPH assay. In brief, 150 µL of Ramalin, its derivatives, and BHA solutions at concentrations of 10, 5, 2.5, and 1 mM in MeOH were mixed with 50 µL of 0.1 mM DPPH in MeOH. The mixtures were then incubated in the dark at room temperature for 30 min. Absorbance was measured at 540 nm. All experiments were conducted in triplicate.

### 4.4. Cytotoxicity and Anti-Inflammation Activity Assays

#### 4.4.1. Cell Culture

RAW 264.7 cells were maintained in Dulbecco’s Modified Eagle Medium (DMEM, Sigma-Aldrich, St. Louis, MO, USA) supplemented with 10% heat-inactivated fetal bovine serum (FBS, Invitrogen, Burlington, ON, Canada) and 1% (*w*/*v*) antibiotic-antimycotic solution (Invitrogen, Grand Island, NY, USA). The cells were cultured in a humidified incubator at 37 °C under 95% air and 5% CO_2_ conditions.

#### 4.4.2. Cytotoxicity Assay

Cell viability was evaluated using the MTT assay (3-(4,5-Dimethyl-2-thiazolyl)-2,5-diphenyl-2H-tetrazolium bromide, Amresco, Solon, OH, USA). RAW 264.7 cells were seeded at a density of 2 × 10^5^ cells/mL in 96-well plates and treated with varying concentrations of Ramalin and its derivatives for 24 h. Subsequently, 5 µL of a 5 mg/mL MTT solution was added, and the cells were incubated for 4 h at 37 °C. Afterward, 100 µL of fresh DMSO was added to dissolve the formazan crystals, and the absorbance was measured at 570 nm using a microplate reader (Thermo Scientific Inc., San Diego, CA, USA). Relative cell viability was determined by comparing absorbance values with the untreated control group. All experiments were performed in triplicate.

#### 4.4.3. Nitric Oxide Production Assay

The production of NO was quantified by measuring nitrite levels in the culture supernatants using the Griess reagent (1% sulfanilamide, 0.1% N-(1-naphthyl)-ethylenediamine dihydrochloride, and 5% phosphoric acid). Briefly, 1 × 10^6^ cells/mL were seeded in 96-well plates and treated with different concentrations of Ramalin and its derivatives for 1 h at 37 °C. The cells were then stimulated with 0.5 μg/mL lipopolysaccharide (LPS, Sigma-Aldrich, CA, USA) for 24 h in a total volume of 200 μL. After incubation, 100 μL of culture supernatant was mixed with an equal volume of Griess reagent and incubated at room temperature for 5 min. Sodium nitrite was used as a standard for calibration, and nitrite concentration was determined by measuring absorbance at 540 nm. All experiments were conducted in triplicate.

#### 4.4.4. NLRP3 ELISA Assay

To assess NLRP3 inflammasome activation, RAW 264.7 macrophages were seeded at a density of 5 × 10^5^ cells/well in 96-well plates. The cells were pre-treated with 20 µM of Ramalin and its derivatives for 1 h, followed by stimulation with 0.5 μg/mL LPS for 24 h. The levels of NLRP3 in the culture supernatant were quantified using a commercially available ELISA kit (Lsbio, WA, USA) according to the manufacturer’s protocol. Absorbance was measured at 450 nm, and NLRP3 concentrations were determined using a standard curve, expressed in ng/mL. To ensure statistical accuracy, all experiments were performed in duplicate.

### 4.5. BACE-1 Activity Assay

The inhibitory activity of Ramalin derivatives against β-secretase (BACE-1) was evaluated using a β-Secretase FRET kit (BACE-1, Thermo Fisher Scientific, San Diego, CA, USA) as previously described [37]. Stock solutions of Ramalin and its derivatives were prepared in deionized distilled water (DDW) at a concentration of 20 mM. Various concentrations of Ramalin and its derivatives were tested in black 96-well microplates, with 10 µL of the BACE-1 substrate added to each well. The reaction was initiated by adding 10 µL of a 3× BACE-1 enzyme solution. Plates were incubated at room temperature for 60 min in the dark. After incubation, 10 µL of 2.5 mM sodium acetate was added to terminate the reaction. Fluorescence was measured using a multimode plate reader (Multiskan™ GO, Thermo Scientific, Waltham, MA, USA) with an excitation wavelength of 545 nm and an emission wavelength of 585 nm. The IC_50_ values were determined by plotting relative fluorescence units per hour (RFU/h) against the logarithmic inhibitor concentrations. All experiments were performed in triplicate.

### 4.6. Anti-Tau Aggregation Inhibition Assay

Tau-BiFC SH-SY5Y were cultured in Dulbecco’s Modified Eagle’s Medium (DMEM, WELGENE, Gyeongsan, Republic of Korea) supplemented with 10% fetal bovine serum (FBS, Gibco, Grand Island, NY, USA) and 1% penicillin/streptomycin (Capricorn Scientific, Ebsdorfergrund, Germany). Cells were maintained at 37 °C in a humidified incubator with 5% CO_2_. For experimental treatment, cells were incubated with 10 μM of Ramalin and its derivatives for 24 h. Following treatment, BiFC fluorescence intensity was measured using the Cytation 5 (BioTek Instruments, Winooski, VT, USA).

### 4.7. Statistical Analysis

Graphs were created and statistical analysis was conducted by using Prism 8 software (GraphPad Software, Boston, MA, USA). Statistical significances was measured using one-way ANOVA with Dunnett’s multiple comparisons test. All data were presented as mean ± standard deviation (SD).

## 5. Conclusions

In this study, we synthesized and characterized a series of novel Ramalin derivatives to evaluate their potential as multi-target therapeutic agents for Alzheimer’s disease. The derivatives exhibited significant antioxidant and anti-inflammatory activities, with RA-2OMe, RA-4OMe, RA-2CF3, and RA-4OCF3 demonstrating potent NO-inhibitory and radical scavenging effects. Moreover, derivatives such as RA-NAP and RA-PYD showed promising BACE-1 inhibitory and anti-tau aggregation activities, positioning them as potential dual-action candidates for AD therapy.

Although in silico predictions indicated limited BBB permeability, these derivatives hold therapeutic promise, warranting further investigations using advanced drug delivery systems to enhance CNS bioavailability. Future in vivo studies will be essential to validate the neuroprotective effects observed in vitro and to explore their pharmacokinetic and pharmacodynamic properties.

These findings provide initial insights into the biological activities of Ramalin derivatives, highlighting their potential relevance in AD treatment strategies, though further research is needed to confirm their clinical applicability.

## Data Availability

Data sharing is not applicable to this article as no new data were created or analyzed in this study.

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
