# Peer review of "Synthesis and Biological Evaluation of Novel Ramalin Derivatives as Multi-Target Agents for Alzheimer’s Disease"

_molecules, 2025, doi:10.3390/molecules30092030_

Round 1

Reviewer 1 Report (Previous Reviewer 3)

Comments and Suggestions for Authors

The manuscript presented here deals with using natural ramalin to prepare derivatives evaluated as multitarget agents for the treatment of Alzheimer's disease. This subject is of great relevance and also of great interest for ​​medicinal chemistry and drug design.

  1. The text should provide more information about ramaline. Other works published by the same group provide a more detailed approach to ramaline. However, the manuscript under analysis lacks information. It must be improved.

  1. The authors should describe which structural attributes of ramaline make it an interesting natural product for the proposed application. This topic should be addressed in more detail in the manuscript.

  1. The authors report in the text that the substituents were chosen due to their physicochemical properties. They suggested modifying the phenyl group or introducing functional groups. What was their basis? Where did the inspiration come from? These molecular design issues should be discussed better, justified, and referenced.

  1. Experimental section: It is necessary that the authors include the melting points of the solid products.

  1. Page 4 (lines 121-123): The authors used Ramalin as an antioxidant standard. Is ramalin a common standard for this type of test?

What is the justification for not using more than one method to evaluate other antioxidant pathways, such as FRAP, ABTS, ORAC, etc.? What is the reason for using only DPPH?

  1. Page 6 (lines 172-183): Some information described in this section of the text differs from what appears in the experimental section, for example, the number of RAW cells, incubation time (after MTT measurement), etc.

  1. Figure 3 (page 8): The authors must include a standard drug for comparison.

  1. Figure 4 (page 9): The authors must include a standard.
  2. Line 329: Correct: Figure 5A instead Figure 1ª.

  1. The authors state (lines 365-366): “Although not all derivatives exhibited strong anti-inflammatory effects, structurally similar compounds consistently demonstrated moderate activity.”   This statement is confusing, and the authors must explain:  What are the “similar compounds”? The authors could improve this statement and also cite the respective bibliographical references.

Author Response

Reviewer 1 comments.

We sincerely thank Reviewer 1 for the thorough and insightful review of our manuscript. We greatly appreciate the constructive comments and helpful suggestions, which have allowed us to significantly improve the clarity, depth, and scientific rigor of our work. Below, we provide point-by-point responses to each of the reviewer’s comments, along with the corresponding revisions made to the manuscript.

Comment 1 & 2

  1. The text should provide more information about ramaline. Other works published by the same group provide a more detailed approach to ramaline. However, the manuscript under analysis lacks information. It must be improved. 
  1. The authors should describe which structural attributes of ramaline make it an interesting natural product for the proposed application. This topic should be addressed in more detail in the manuscript.

 Response 1 & 2:

 We thank the reviewer for these insightful comments. In response, we have significantly revised the Introduction section to provide a more comprehensive overview of Ramalin. Specifically, we elaborated on its natural origin (Ramalina terebrata), unique structural features—namely its glutamic acid backbone, hydrazine linker, and phenyl group—and their relevance to biological activity. We also emphasized Ramalin’s structural simplicity and high modifiability, which enable the introduction of diverse functional groups that significantly influence its antioxidant, anti-inflammatory, BACE-1 inhibitory, and anti-tau aggregation properties. These revisions clarify why Ramalin is considered a chemically versatile and pharmacologically promising scaffold for multi-target drug development. The revised content is located in the middle portion of the Introduction and is supported by additional references. (Line 87-102)

Comment 3.

 The authors report in the text that the substituents were chosen due to their physicochemical properties. They suggested modifying the phenyl group or introducing functional groups. What was their basis? Where did the inspiration come from? These molecular design issues should be discussed better, justified, and referenced.

 Response 3:

 Thank you for your valuable suggestion. In response, we have revised the Introduction to clarify the rationale behind our molecular design. In our previous studies, we observed that changes in the substituent group on the Ramalin scaffold led to significant differences in biological activities. Notably, the position and type of the substituent appeared to exert a greater influence than their electronegativity alone. Based on these observations, the current study was designed to systematically investigate how modifying the functional groups and phenyl ring structure affects antioxidant, anti-inflammatory, BACE-1 inhibitory, and anti-tau aggregation activities.

 Moreover, the multifactorial nature of Alzheimer’s disease—characterized by oxidative stress, neuroinflammation, Aβ accumulation, and tau aggregation—makes it extremely challenging to develop molecules that can simultaneously address all pathological pathways. Therefore, we focused on structurally modifying a known multi-target scaffold, Ramalin, as a practical and rational starting point for derivative synthesis. This strategy allowed us to explore how subtle structural changes could enhance or diversify biological activities. The revised explanation appears in the fourth paragraph on the introduction (Line 112 - 115)

Comment 4.

 Experimental section: It is necessary that the authors include the melting points of the solid products.

 Response 4:

 Thank you for your valuable comment. Due to time constraints during the revision process, we were unable to measure and include the melting points of the synthesized solid compounds. In our previously published studies on Ramalin derivatives submitted to Molecules, melting point data were not included, and as such, we initially did not prioritize their measurement. We sincerely apologize for this oversight. If acceptable to the reviewer, we plan to measure and report the melting points of the active compounds in a future publication that further explores their structure–activity relationships and physicochemical properties. We appreciate your understanding.

Comment 5.

 Page 4 (lines 121-123): The authors used Ramalin as an antioxidant standard. Is ramalin a common standard for this type of test?

  What is the justification for not using more than one method to evaluate other antioxidant pathways, such as FRAP, ABTS, ORAC, etc.? What is the reason for using only DPPH?

 Response 5:

 Thank you for your insightful comment. In our study, Ramalin was used as the antioxidant reference compound, not as a general standard, but rather because the purpose of the study was to identify derivatives with equal or superior antioxidant activity compared to Ramalin itself. The DPPH assay was chosen as a simple and widely used method to determine whether the synthesized compounds retained antioxidant activity similar to or better than Ramalin.

 We acknowledge that evaluating antioxidant capacity using multiple methods (such as FRAP, ABTS, ORAC) would provide a more comprehensive understanding of antioxidant mechanisms. However, antioxidant activity was not the primary focus of this study, and the DPPH assay was employed as a preliminary screening tool. In our previous publication on Ramalin [Reference 33], where antioxidant activity was the main subject, we employed a more extensive assessment to characterize its antioxidant profile.

 Should any derivative show particularly strong or novel antioxidant properties, we agree that further investigations using additional methods like FRAP, ABTS, or ORAC will be necessary. We sincerely appreciate your suggestion and will take it into account for future studies.

Comment 6.

Page 6 (lines 172-183): Some information described in this section of the text differs from what appears in the experimental section, for example, the number of RAW cells, incubation time (after MTT measurement), etc.

Response 6:

Thank you for your careful observation. We have confirmed that there was an inconsistency between the Results and the Materials & Methods sections regarding the number of RAW 264.7 cells and the incubation time after MTT treatment. The information provided in the Materials & Methods section is correct, and the discrepancies in the Results section have been revised accordingly to ensure consistency. We appreciate your attention to detail.

Comment 7 & 8.

  1. Figure 3 (page 8): The authors must include a standard drug for comparison.
  2. Figure 4 (page 9): The authors must include a standard.

Response 7 & 8:

 Thank you for your valuable comments. In both the NO production and NLRP3 inhibition assays, we used Ramalin as the reference compound for comparison. While we agree that including a well-established standard drug would provide more robust benchmarking, these experiments were conducted as part of a preliminary screening effort aimed at identifying Ramalin derivatives with improved or comparable activity. Based on our previous studies, Ramalin has consistently shown reproducible biological activity and was therefore used as a practical in-house standard in this context.

 We fully acknowledge the importance of using established reference drugs and will include appropriate standards in follow-up studies where more comprehensive validation is performed. We appreciate your suggestion and will apply it in future experimental designs.

Comment 9.

 Line 329: Correct: Figure 5A instead Figure 1ª.

Response 9:

 Thank you for pointing this out. We have corrected the typographical error as suggested and changed “Figure 1A” to “Figure 5A” in line 329. We appreciate your attention to detail.

Comment 10.

 The authors state (lines 365-366): “Although not all derivatives exhibited strong anti-inflammatory effects, structurally similar compounds consistently demonstrated moderate activity.”   This statement is confusing, and the authors must explain:  What are the “similar compounds”? The authors could improve this statement and also cite the respective bibliographical references.

Response 10:

 Thank you for pointing out this ambiguity. We agree that the statement “structurally similar compounds” was vague and have revised the sentence to clarify its meaning. Specifically, we referred to the group of derivatives containing a hydrazinophenyl moiety—including RA-2OMe, RA-4OMe, RA-2CF3, RA-3CF3, and RA-4OCF3—which share structural similarity with the parent compound Ramalin. These compounds consistently demonstrated moderate to strong anti-inflammatory activity, particularly in NO and NLRP3 inhibition assays.

 We have rephrased the sentence to improve clarity and have also included the appropriate references to our previous studies [35, 36], where the structure–activity relationship of this scaffold was explored. The revised sentence appears in the Discussion section (line 384 - 388).

 We sincerely appreciate Reviewer 1’s thoughtful and constructive comments. Your feedback has been instrumental in improving the clarity and scientific rigor of our manuscript. We have carefully addressed each point and revised the manuscript accordingly.

 Thank you once again for your valuable insights.

Sincerely,

Joung Han Yim

Division of Polar Life Sciences, Korea Polar Research Institute

Incheon 21990, Korea

Phone number: +82-32-760-5540

Fax number: +82-32-760-5509

jhyim@kopri.re.kr

Reviewer 2 Report (New Reviewer)

Comments and Suggestions for Authors

The manuscript describes the synthesis of novel Ramalin derivatives and their evaluation as antioxidant, anti-inflammatory, and neuroprotective agents for the treatment of AD. The obtained results are noteworthy and presented clearly. The manuscript is well-written, thus it deserves publication. Nevertheless, some modifications/integrations are needed before the acceptance.

Abstract, line 32. Please, rephrase  “….with IC50 values superior to positive control”. It is incorrect because the IC50 values of the new compounds are lower than the positive control.

Figure 1. The structure of Ramalin should be inserted

Scheme 1. The compounds, both intermediates p-Glu-Hyd and the final RA-derivatives, should be numbered.

Experimental section, 4.2.1. The General method for the synthesis of p-Glu-Hyd analogues is described, but the data (Yields, 1H NMR spectra, etc etc) of these intermediates are completely lacking. The authors must insert these informations.

Page 15, In the “Antioxidant Activity Assay” section we read “Absorbance was measured at 540 nM”. I suppose it should be “517 nM”.

Author Response

Reviewer 2 comments

We sincerely thank Reviewer 2 for the thoughtful and constructive comments provided. Your insightful suggestions have been extremely helpful in refining the clarity and scientific rigor of our manuscript. Below, we provide detailed responses to each of your comments and describe the corresponding revisions made to the manuscript.

Comment 1

 Abstract, line 32. Please, rephrase  “….with IC50 values superior to positive control”. It is incorrect because the IC50 values of the new compounds are lower than the positive control.

Response 1:

 Thank you for your helpful comment. We agree with the reviewer that the phrase “IC50 values superior to positive control” may be misleading, as a lower IC50 value indicates stronger inhibitory activity. Accordingly, we have revised the sentence in the abstract to:
“…with IC50 values lower than that of the positive control, indicating greater inhibitory potency.”
We appreciate your attention to this important detail and have updated the manuscript accordingly.

Comment 2

 Figure 1. The structure of Ramalin should be inserted

Response 2:

 Thank you for your suggestion. As recommended, we have inserted the chemical structure of Ramalin into Figure 1 for clarity and completeness.

Comment 3

 Scheme 1. The compounds, both intermediates p-Glu-Hyd and the final RA-derivatives, should be numbered.

Response 3:

 Thank you for your suggestion. We agree that numbering the intermediates (p-Glu-Hyd) and final RA-derivatives in Scheme 1 could improve clarity. However, if permitted, we would prefer to retain the current presentation style, which is consistent with our previously published work on Ramalin derivatives. To maintain continuity and avoid confusion when comparing with prior literature, we have not modified Scheme 1 in this revision. That said, if the reviewer considers compound numbering essential, we will be happy to revise the figure accordingly in the next round. We appreciate your kind understanding.

Comment 4

 Experimental section, 4.2.1. The General method for the synthesis of p-Glu-Hyd analogues is described, but the data (Yields, 1H NMR spectra, etc etc) of these intermediates are completely lacking. The authors must insert these informations.

Response 4:

 Thank you for your helpful comment. The yields of the p-Glu-Hyd intermediates were included in the overall yield calculation for the final RA-derivatives and were therefore not separately reported. In our previous three publications on Ramalin derivatives (References 35, 36, and 37), similar experimental procedures were used, and no specific concerns were raised regarding intermediate characterization. As such, we proceeded directly to the deprotection step after confirming purity and yield, without collecting separate analytical data for the intermediates.

 Unfortunately, due to time constraints and the need to resynthesize the intermediates, we were unable to provide this information during the current revision. However, we fully acknowledge the reviewer’s point and will reflect this feedback in future derivative synthesis to ensure full reporting of intermediate data. We kindly ask for your understanding.

Comment 5

 Page 15, In the “Antioxidant Activity Assay” section we read “Absorbance was measured at 540 nM”. I suppose it should be “517 nM”.

Response 5:

 Thank you for your accurate observation. We fully understand that DPPH assays are conventionally measured at 517 nm. However, in our case, absorbance was measured at 540 nm because the Multiskan™ GO microplate reader (Thermo Scientific) used in our laboratory does not support direct measurement at 517 nm. Among the available wavelengths, 540 nm was the closest to 517 nm and was therefore selected.

 To validate this approach, we performed a comparative analysis using a separate instrument capable of measuring at 517 nm. The results showed that the antioxidant activity trends were consistent across both wavelengths. Based on this validation, we proceeded with 540 nm measurements throughout our study and have clearly indicated this in the manuscript. We appreciate your thoughtful comment and the opportunity to clarify this point.

 We once again appreciate Reviewer 2’s time and effort in reviewing our manuscript. Your comments have greatly contributed to improving the overall quality of our work. While we have addressed all points to the best of our ability, we kindly ask for your understanding regarding the few items we were unable to revise due to practical limitations, such as time constraints or the need for re-synthesis. We will carefully consider and incorporate your suggestions in future studies.

Thank you again for your constructive input.

Sincerely,

Joung Han Yim

Division of Polar Life Sciences, Korea Polar Research Institute

Incheon 21990, Korea

Phone number: +82-32-760-5540

Fax number: +82-32-760-5509

jhyim@kopri.re.kr

This manuscript is a resubmission of an earlier submission. The following is a list of the peer review reports and author responses from that submission.

Round 1

Reviewer 1 Report

Comments and Suggestions for Authors

The manuscript describes quite detailed study of novel Ramalin derivatives on spectrum of biotargets involved in pathogenesis of number neurodegenerative diseases (ND). The authors synthesized about dozen of new Ramalin derivatives and analyzed structure activity relationships in relation to the ability of these compounds to decrease an oxidative stress, cholinesterase inhibition, NO-generating activity, anti-inflammation property. As the main result of this study it could be outlines the determination of structural fragments that play crucial role in each mechanism of action and as a total result the revealing of lead-compounds that show optimal multifunction action on the selected group of biotargets.

On my point of view there are only some minor suggestions that the authors should correct in the manuscript:

  1. As the main goal of this manuscript is to describe the novel potentially multitarget agents for the treatment of ND, in Introduction the authors should list more published papers in this field (in particular, such article as the review of Makhaeva et al. Chem.-Biol. Interact., 2019; review of Xingyou Mo et al. Molecules 2024, etc.).
  2. When the authors present results of Ramalin derivatives on specific targets and processes, in each case they should compare it with the parent compound - Ramalin, in particular in characterizing BuChE inhibition, on the proliferation of RAW264.7 cells., and so on.. 

Reviewer 2 Report

Comments and Suggestions for Authors

The submitted manuscript entitled “Synthesis, Biological Evaluation of Novel Ramalin Derivatives as Multi-Target Agents for Alzheimer’s Disease” follows the pattern of several ramalin chemical optimization strategies developed by same authors in recent years to improve its neuroprotective properties. In this case, a small library of aryl/alkyl analogues was synthesized and evaluated for their antioxidant, anti-inflammatory and antiaggregant properties. The manuscript is pleasant to read with a step-by step well described biological characterization. Below some critical issues that has to be addressed before publication:

  • In this case, for reason of clarity, these derivatives should be better defined as multifunctional or pleiotropic due to the ability in modulating different cellular processes more than specific targets. Furthermore, multi-target agents should be developed with the aim to identify a balanced compound in the resulting biological properties, while in this case NLRP3 inhibition has been evaluated at 40 uM, tau aggregation at 10 uM and for BACE-1 resulted active in low micromolar range. Therefore, in addition to justify the differences in the exploited concentration, in the discussion this fact should be taken into account.
  • For reasons reported above and due to the limits emerged in the experimental outputs, every consideration in abstract, discussion and conclusion that refer to the “promising potential as multi-target therapeutic agents” or “candidates for AD” of these derivatives should be reasonably soften.
  • In the antioxidant evaluation it is missing the statical analysis as well as number of replicates carried out.
  • In every biological investigation ramalin should be used as inner reference to compare the contribution of newly introduced chemical modifications, while this was done only for antioxidant effect and tau aggregation evaluation. Otherwise, the reported results should refer to previous findings with ramalin or ramalin analogues.
  • In paragraph 2.1 there is a duplicated part describing RA-NAP synthesis
  • In 4.2.1 “p-Glu-Hyd” instead of “p-Glu-Bn” analogues as they are called along the manuscript

Reviewer 3 Report

Comments and Suggestions for Authors

This referee does not see any significant results in the data set presented by the authors that would justify the publication of another article with the same focus and no improvement over the results from four years ago (Molecules 2021, 26, 6445). 

Find some comments in the attached PDF.

Reviewer 4 Report

Comments and Suggestions for Authors

Referee Report

Ms ID: molecules-3502075

The Ms entitled”Synthesis, Biological Evaluation of Novel Ramalin Derivatives as Multi-Target Agents for Alzheimer’s Disease” appears as a multi-disciplinary work in the area of drug development for neurodegenerative diseases. Although it appears as a continuation of previous work recently reported by the same authors on Ramalin derivatives and so not really new in this area, globally it appears as a good work. Therefore although presenting some deficits, in my opinion it can be accepted for publication in this journal after a minor revision, namely after addressing the following comments and questions.

Comments

-Introduction

 Reference 10 is not adequate within the context it is used, and so it should be substituted.

- Results and experimental

The authors should include the synthetic procedures and its schematic representation (in Scheme1) for the preparation of all intermediates, including the hydrazine-substituted derivatives.

- The authors should make clearer for the readers the option of the RAW 264.7 cell line for the toxicity evaluation (instead of the usual neuroblastome cell line).